# Spatiotemporal Variability in the Hydrological Regimes and Water Resources of the Ouham River Basin at Batangafo, Central African Republic

**Cyriaque Rufin Nguimalet** [1,*] and **Didier Orange** [2]

1   Département de Géographie, Faculté des Lettres et Sciences Humaines,
    Bangui BP 1037, Central African Republic
2   UMR 5151- HSM, HydroSciences Montpellier, IRD, UM, CNRS, 34090 Montpellier, France;
    didier.orange@ird.fr
*   Correspondence: cyriaque.nguimalet@univ-bangui.org

**Abstract:** This paper examines the effect of rainfall decline on water resources in each sub-basin (Bozoum: 8100 km$^2$ and Bossangoa: 22,800 km$^2$) and at the outlet of Batangafo (43,650 km$^2$) over the 1951–1995 period, due to a lack of measurements since 1996. Annual, monthly, and daily series of rainfall and discharges were subjected to statistical tests (rainfall and flow indices, SPI, search for ruptures/breaks, depletion coefficient, and potential groundwater discharge) to present and discuss the rainfall variability impact on the water resources of the whole basin. The average rainfall per sub-basin decreases from the west to the east according to the Ouham river direction: 1423 mm at Bozoum, 1439 mm at Bossangoa, and 1393 mm at Batangafo, the main outlet. The SPI approach provides evidence of a moderate to normal drought in the whole basin in the 1980s, mainly compared to the 1970s. Thus, deficient breaks in the rainfall series of the Ouham Basin at Batangafo were noticed in 1967 (Bossangoa and Batangafo) and 1969 (Bozoum). A declining rainfall of −5% on average tended to have the highest impact on the runoff deficit, from about −30 to −43%. The deficit seems more important from west to east, and is also high over the groundwater in each outlet (−33% at Bozoum, −29% at Bossangoa, and −31% at Batangafo) in the 1986–1995 period, despite rainfall recovery in 1991 having generated a flow increase in 1995 at Bossangoa as well as at Batangafo. At the same time, Chari/Logone at Ndjamena recorded critical discharges in both 1987 (313 m$^3$/s) and 1990 (390 m$^3$/s) before they increased, such as on the Ouham. These results demonstrate the decline in water resources in the Ouham River, and their direct impact on the water level of the Chari River and Lake Chad in the targeted period.

**Keywords:** spatiotemporal variability; hydrological regimes; water resources; Ouham at Batangafo; Bozoum; Bossangoa; Central African Republic

## 1. Introduction

Since the mid-20th century, land use change has been considered the main driver of the hydrological evolution of soils and watersheds [1–6], with numerous impacts on ecosystem services [3,7] and livelihoods [8]. But with the current issue of climate change, the concept of non-stationarity in the relationship between the rainfall regime and hydrological behavior is questioned more and more [9,10], without any clear tendency due to the integration of human and natural systems [11–14]. Both interacting human and natural systems can affect water resource distribution anywhere, inducing livelihood degradation and conflicts, as underlined in the Lake Chad Basin (LCB) [15].

The LCB is an endorheic basin that was and still remains a place of human crossroads, despite the drastic depletion of its water resources [16]. Lake Chad covered 25,000 km$^2$ in the 1960s, reaching only 3000 km$^2$ at the end of the 1990s. And since the resumption of pluviometric indices in the 2000s, the size of Lake Chad has remained particularly small

and gives rise to fears of support for political instability in the area [15,17]. It creates an urgency in this very political and environmentally sensitive geographic area to highlight the relationship between the impact of humans and regional freshwater systems to ensure successful water management in the future [18].

In many places in Africa, the river drought induced by the rainfall pejoration in the 1970s has not ended in spite of the recent increases in rainfall [19]. In the Sudanian area, stream flows have decreased much more than the rainfall pejoration, although the runoff coefficients have increased in the Sahelian area [20,21], induced by changes in the relationship between hydrological parameters, as it has been demonstrated in West Africa with the Sahelian Paradox [22] and in Central Africa within the Congo Basin [23–25]. However, ref. [26] has demonstrated that the hydrological functioning of the Ubangi River, the northern part of the Congo Basin in the Central African Republic (CAR), has not changed in spite of the climate disruption due to the rainfall break that occurred in 1970. This is attributed to the low human impact on this basin and to the small change in the vegetation cover. Despite the exceptional duration of the drought period (1970–2009), currently, the slightest recovery in rainfall volumes results in an immediate contribution from the Ubangi River Basin aquifer, presenting an aquifer that is still reactive [26].

The Lake Chad Basin (LCB), between the Sahelian and Sudanian climate zones, is on the northern border of the Ubangi River Basin. A recent study [27] confirmed that the highest correlation able to explain the Lake Chad surface was with the monthly flows of the Chari–Logone River system [28,29]. The Ouham River, which drains the northern part of the CAR, is the most important tributary of the Chari River, feeding Lake Chad [30]. The Ouham–Bahr Sara–Chari is a river of 1590 km in length that originates at 700 km in the western part of the CAR over the Bouar–Baboua Plateau (1120 m) [31]. As has been well documented [30], rainfall deficits have been strongly marked over the Ouham River Basin since 1969–1970. However, the hydrological dynamics of the Ouham River have not yet been studied. Nevertheless, it has been reputed that the lowest water levels of Lake Chad could be linked to the impact of the decline in the hydrological regime in this region, which started in 1970 [21,23,26,32–40]. In the whole LCB, this also implies the same evolution of its tributaries as the Ouham River. According to [41], "the Bahr Sara is the main tributary branch of the Chari as well on the flowed volume as on risings or low-water levels". Indeed, ref. [42] defined two contrasted climatic episodes, wet (1960–1971) and dry (1982–1997), over the Chari–Logone, suggesting a respective rainfall contribution of 75% and 15% to the runoff. This drought episode corresponds to the sequence of severity recorded in the 1980s as well, on the large basins and rivers of the region [21,34,38,43], as well as on the small rivers or elementary basins [44–46]. This phenomenon is led by the rainfall decline (few and/or high), and the rate varies from one area to another. According to [47], this declining rainfall context affects the water course hydraulicity in intertropical Africa, which reduces water resources over the Sahel to Central Africa. The authors of [48] noticed that consecutive droughts generated the isohyet descent from north to south for about thirty years, impacting the livestock transhumance according to this gradient. The authors of [18] observed that, during the 1970s and 1980s high drought episodes, these isohyets migrated about 120 km toward the south compared to their position in the wet period (1951–1969). This north–south gradient was observed by [49] when he pointed out the earliness of the second hydroclimatic period (1955) in the Ubangi Basin in the north compared to that of Congo to the south (1960).

This study analyzes the interannual evolution of the rainfall–discharge relationship at three existing hydrological gauging stations of the Ouham River, from upstream to downstream at the outlets of Bozoum, Bossangoa, and Batangafo (Figure 1). Thus, we aim to evidence the effect of rainfall decline on the reduction in water resources at these three hydrological sub-systems. What is the spatiotemporal evolution of the rainfall (P) and discharge (Q) regimes regarding drought persistence in the Ouham River Basin? Which have an impact on the water resources of the Ouham River Basin, and what are the consequences for Lake Chad? This study of the Ouham River Basin also aims to represent a geographical

transition between Sahelian and Sudanian climate zones, i.e., between wet and dry Africa. The hypothesis is that the evolution of the water resources of the Ouham River Basin from 1951 to 1995 directly explains the water level of Lake Chad, as the main contributor to the Chari/Logone River. The method used involves characterizing the impact of rainfall and discharge in terms of breaks and homogeneous hydroclimatic periods. This work highlights the importance of the hydrological regime of the Ouham River in controlling the Lake Chad water level. At last, an extrapolation of the homogeneous hydroclimatic periods recently recorded in the Ubangi River Basin on the southern border of the Ouham River Basin is discussed to assess the synchronic effect of the current hydrological regime in both basins of the CAR (in part of the Lake Chad Basin and in part in the Congo River Basin), in order to evaluate the importance of drought in the Region.

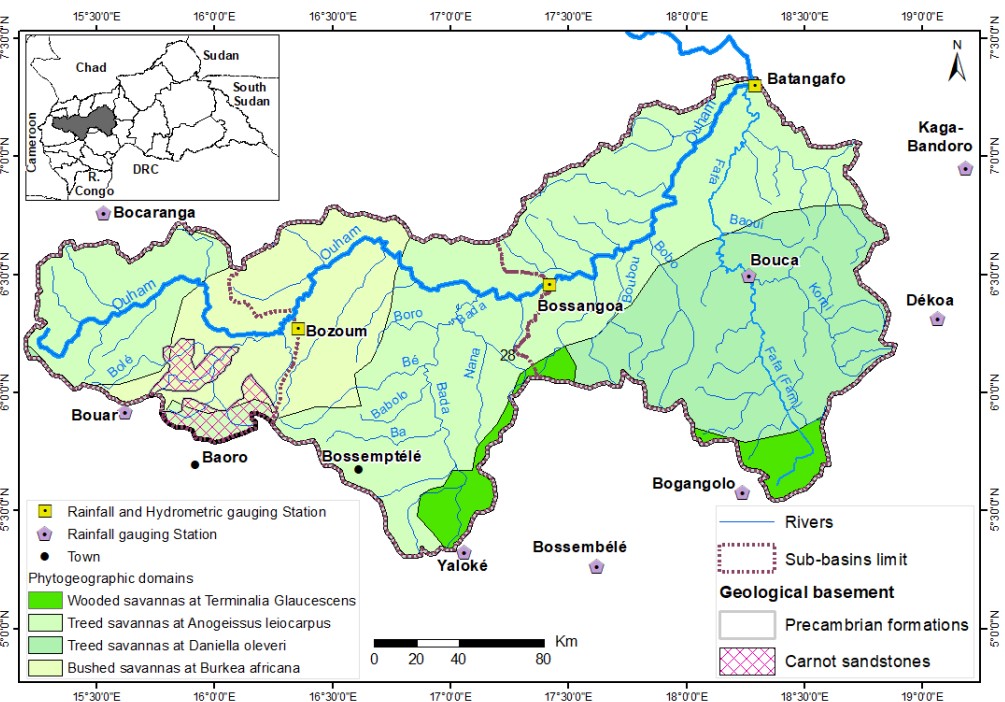

**Figure 1.** The Ouham River Basin at Batangafo, with sub-catchments at Bozoum and Bossangoa, modified from [30], and the rainfall and hydrometric gauging stations.

## 2. Data and Methods

### 2.1. The Ouham River Basin and Its Sub-Basins

The South limit of the Ouham River Basin coincides with the interfluve Chad–Congo. The Ouham River Basin is delimited by the Ubangi and the Sangha Basins in the South of the Lake Chad Basin, in the CAR (Figure 1). The Ouham River flows from the West to the East. The hydrometric gauging station of Bozoum controls the upper part of the basin at an elevation of 629 m (6°20′ North–16°21′ East, kilometric point or kp 200) [50], and was installed on 8 March 1952 with a sub-basin of 8100 km². The outlet of Bossangoa (22,800 km²) is on the middle stream at 445 m [50], and was installed at the end of May 1951 under the bridge of the Bossembélé–Bossangoa road (6°28′ North–17°27′ East), at kp 368 [31]. The hydrometric gauging station at Batangafo (7°43′ North–19°59′ East) controls the whole basin; it is situated at 3945 m [50], at kp 550, and was installed on 10 April 1951 [31], immediately downstream of the ferry on the right bank, with a basin area of 43,650 km².

The basin relief comprises three topographic unities: the mountains (up to 1000 m), which are the prolongation of the Cameroonian Adamaoua, the plateaus (1000–500 m), and the plains (500–300 m). The elevations vary from 1190 m (Niem at the West of the basin) to 431 m (alluvial plain in the Batangafo area) [30]. The Ouham River originates near

Niem, on a small escarpment (cote of 1118 m: $6°15'$ N–$15°19'$ E) of the connecting surface of Mount *Lalengué/Lélengué*, which was identified as a group of granitic boulders by the Commandant Lenfant in 1907 [31]. The plateaus dominate in the Ouham upper basin and partially in the middle basin up to Bossangoa, and also in the sub-basin (600–500 m) of the Fafa River feeding the Ouham at Batangafo just up to the hydrometric station. As for the plains, they spread out from the Ouham valley up to Béa-bac ($6°29'$ North–$17°05'$ East; cote 485.80 m), and the Baba River, with tributaries on the right bank, where they are very narrow until Bossangoa. From there until Bouca, and afterward Batangafo, these plains are wide towards the Bahr–Sara at the north.

The basin substratum is built up by the Precambrian basement, which comprises a discordant cover formation or superficial deposits [51]. The Precambrian basement comprises the "Complexe de base", composed of crystalline rocks (circumscribed granites and massifs) and metamorphic facies (charnockite, gneiss, and quartzite). The sedimentary covers are constituted of Mesozoic sandstones (fluvio-lacustrine) of Carnot in the southwest basin, of the Continental Terminal (paleo and neo-Chadian geological stages), and of the recent alluvium [52,53].

The CAR shield forming the Chadian upper basin of the Ouham River presents storied steps steered to the West–East until Batangafo. This relief influences the irregularity of the river's longitudinal profile, noticeable due to the principal morphological characteristics of its channel with sequences of rapids (irregular slope or rocky high-bed). Some alternations between the rectilinear and sinuous channels are observed. Then, the Ouham River is meandriform, with a sinuosity coefficient of around 2 [30].

In this basin, three of the CAR Sudano-climate variants are present from South to North (Ubangian, Guinean, and Sahelian). The Sudano–Guinean prevails over three-quarters of the total basin area, with both contrasted seasons (rainy and dry) and a variable annual mean rainfall of 1200 to 1500 mm. Consequently, two phytogeographic domains are also distinguished from South to North: the Sudano–Guinean and the medio-Sudanian [54]. The first is composed of dry dense forests and Savannah mainly occupied by *Lophira lanceolata*, and the second domain is marked by *Uapaca togoensis* and *Vitellaria paradoxa*. One-quarter of the basin has a Sahelian climate.

Human activities are the primary economy: itinerant agriculture, artisanal mining extraction, and wood and charcoal production, etc. According to [30], the North-West and North regions, comprising the studied Ouham River Basin, are the areas with the most inhabitants. Indeed, the population increased from 741,871 inhabitants in 1988 [55] to 1,474,179 inhabitants in 2003, for a density of 8.8 inhabitants/km$^2$; this accounts for 41% of the national population [56]. This mainly attests to an increase of 100% in 15 years, corresponding to an exceptional annual increase of 6.5%. These human densities demonstrate the importance of human activities in the Ouham River Basin. Thus, the regions of Ombella-Mpoko (Yaloké), Ouham, and Ouham–Pendé (Figure 1) are reputed to have high agro-pastoralist activities (cultivation of cotton and cereal, and itinerant breeding of bovine). The crop management practice of slash-and-burn and the seasonal damage caused by bushfires to the ecosystems accelerate vegetal and soil degradation. These processes generate surface runoff (Horton-type runoff), leading to water losses in the whole basin.

*2.2. Data and Methods*

2.2.1. Rainfall and Flow Data

A monthly and annual rainfall data series were built from the gauging stations distributed inside and outside the Ouham Basin: Batangafo (1951–1987), Bocaranga (1951–1980), Bossangoa (1951–1995), Bossembélé (1951–1995), Bouar (1951–1995), Bouca (1951–1990), Bozoum (1951–1990) and Yaloké (1951–1980) (Figure 1). They were completed using the rainfall data obtained from the gauging stations of Bogangolo (1953–1987), Dékoa (1953–1986), and Kaga-Bandoro (1951–1995) to complete the rainfall calculation at Batangafo. The elevations of the rainfall stations were as follows: Bocaranga (1072 m), Bouar (1020 m), Yaloké (748 m), Bossembélé (673 m), and Bozoum (672 m). The rainfall stations with the relatively lowest

elevations were Bossangoa (465 m), Bogangolo (587 m), Dékoa (550 m), Bouca (458 m), Batangafo (431 m), and Kaga-Bandoro (410 m). Of the data obtained from these stations, some were used for the calculation of the mean rainfall in the respective sub-basins at Bozoum (Bocaranga, Bozoum, and Bouar stations) and Bossangoa (Bocaranga, Bossangoa, Bozoum, Bossembélé, Yaloké, and Bouar stations). The annual mean and daily mean Q data of the Ouham River were calculated from the hydrometric gauging stations situated at Bozoum (629.22 m), Bossangoa (445.48 m), and Batangafo (394.83 m) over the 1951–1995 period (44 years). All raw daily data were provided by the Institut de Recherche pour le Développement (IRD or ex-ORSTOM: http://www.hydrosciences.fr/sierem, accessed on 4–21 July 2016) and the ASECNA (Agence pour la Sécurité de la Navigation Aérienne en Afrique et Madagascar) at Bangui. Supplementary data were collected in the Annuaires Hydrologiques of the CAR over the 1985–1995 period [57–61].

### 2.2.2. About Data Quality

The quality control of these gathered data consisted of a preliminary analysis to disclose some eventual typing errors and/or a lack of measures in the series (Table 1a for rainfall, Table 1b for discharge). The aim was to evaluate the coherence of the database. This data quality control was conducted to distribute them in valid or continual sequences, and for missing data (some months or years missing according to the considered time scale), fill the gaps, if possible. This is detailed in Table 1.

**Table 1.** (**a**) Rainfall data missing (P) over the Ouham River Basin. (**b**) Discharge data missing (Q) for the Ouham River at the Bozoum, Bossangoa, and Batangafo stations.

| (a) | | | | |
|---|---|---|---|---|
| **Rainfall Stations** | **Period** | **Missing Monthly** | **Missing Years** | **Missing Periods** |
| Batangafo | 1951–1987 (36 years) | 38 months (9%) | 4.2 years (12%) | 1981–1984, 1985 (January and February) |
| Bocaranga | 1951–1980 (29 years) | None (0%) | None (0%) | - |
| Bossangoa | 1951–1995 (44 years) | None (0%) | None (0%) | - |
| Bossembélé | 1951–1995 (44 years) | 7 months (1.3%) | 0.7 year (1.3%) | 1983 (January–April and August–October) |
| Bouar | 1951–1995 (44 years) | None (0%) | None (0%) | - |
| Bouca | 1951–1990 (39 years) | None (0%) | None (0%) | - |
| Bozoum | 1951–1990 (39 years) | 12 months (2.6%) | 1 year (2.6%) | 1986 (January–December) |
| Yaloké | 1951–1980 (29 years) | None (0%) | None (0%) | - |
| Bogangolo | 1953–1987 (34 years) | 12 months (2.6%) | 1 year (2.6%) | 1983 (January–March), 1984 (January, February, August and December), 1985 (February, November and December), and 1986 (September and November) |
| Dékoa | 1953–1986 (33 years) | 8 months (2%) | 0.8 year (1%) | 1981 (January–August) |
| Kaga-Bandoro | 1951–1995 (44 years) | None (0%) | None (0%) | - |
| (b) | | | | |
| **Hydrometric Stations** | **Period** | **Missing Daily** | **Missing Monthly** | **Missing Annual** | **Missing Periods** |
| Ouham at Bozoum | 1951–1994 (43 years, 516 months) | 1951–1952 1964–1965 1976–1986 1994–1995 | 144 months (28%) | 12 completed years (2.33%) | 1951–1952 1964–1965 1976–1986 |
| Ouham at Bossangoa | 1951–1995 (44 years, 528 months) | 1961–1962 1980–1986 1994–1995 | 130 months (25%) | 6 completed years + 4.8 years partially (1%) | 1961–1962 1980–1986 1994–1995 |
| Ouham at Batangafo | 1951–1995 (44 years, 528 months) | 1962–1964 1978–1984 1991–1992 | 177 months (34%) | 10 completed years + 4.8 years partially (1%) | 1962–1964 1978–1984 1991–1992 |

For short periods of missing rainfall and flow data, the monthly reconstitution of the month *i* was performed using the averaging method of months *i* for the two years surrounding the missing data, or by using linear regression/extrapolation. Otherwise, long periods of missing data were filled using data from a nearby hydrological station, upstream or downstream [62,63]. Thus, over the rainfall series, some weak gaps in the three stations (Batangafo, Bozoum, and Bossembélé), respectively, reached 12%, 3%, and 1.33%, according to their series length. As for the different outlets of the Ouham, the filled missing data, respectively, were 28% at Bozoum, 25% at Bossangoa, 34% at Batangafo, and 16% on the Fafa river at Bouca, a right bank tributary supplying the Ouham up to its outlet at Batangafo (Table 1b). On a daily time scale, missing data were globally recorded over the 1961–1965, 1978–1984, 1991–1992, and 1994–1995 periods. Due to their availability, data were used to calculate the depletion coefficients and water volume of the aquifer in each outlet.

All P and Q data should have a Gaussian distribution (Shapiro test, $P > 0.05$). Thus, to appreciate their coherence or evolution regarding wrong/missing data, the Student's *t*-test and Pearson linear correlation were used to compare the means of the two groups and to correlate the P and Q couple, respectively. The probability and significance level of all tests were fixed at 0.05. No difference was established between the rainfall averages of Bozoum (1423 mm) and Bossangoa (1439 mm, $P = 0.64$), between those of Bossangoa and Batangafo (1393 mm, $P = 0.64$), or between those of Batangafo and Bozoum ($P = 0.4$).

The Pearson coefficient (comprised between −1 and 1) is used to detect the presence of a linear relation or to assess/measure the linear dependency between two variables. The Pearson coefficient was found to be between 0.89 and 0.92 for rainfall, between 0.60 and 0.84 for the annual mean discharges, and also near 1, showing a high data dependency between the stations with a positive correlation.

### 2.2.3. Statistical Tests

Before treating the P and Q variables with statistical tests, the R and Q data were disposed of according to the hydrological year, which starts on April 1st, before being treated in the region. At the annual scale, the average rainfall was calculated using the Thiessen method for the whole basin [64,65], even though the simple average of the data per rainfall gauging station was used for the sub-basins of Bozoum and Bossangoa. Both approaches gave similar results, with +1 mm for the Thiessen method. The rainfall index (annual rainfall divided by interannual average of the studied series) was also calculated to compare the evolution of wet and dry years around the 1967–1970s, which show a major climatic break in this studied basin. In addition, the Standardized Precipitation Index (SPI) was calculated over the Ouham River Basin. The data series were adjusted to a Gaussian probability distribution, and then the averaged SPI was equal to 0 [66]. The SPI-positive values indicate rainfall above the medium and the negative values below the median. Thus, drought starts when $SPI \leq -1$, and it ends when its value is positive [67,68]. The calculation of the SPI with the aim of assessing the drought periods was performed using SPEI and haven R packages under the interface Rstudio, fitting the data focused on the time scale of 12 to a Gaussian distribution [69]. At a monthly scale, the data per station simply were averaged to analyze the mean and annual rainfall regimes per sub-basin and for the whole basin, with the aim of evaluating the average reduction before and after the 1970s. In addition, the annual mean Q was used to calculate the flow index (ratio of the annual mean Q to the interannual average) in order to look for the hydrologically wet and dry years per outlet. The evolution of monthly mean Q was also analyzed between the earlier and later periods of 1970.

Break tests (Rank, Buishand, Pettitt, Lee and Heghinian, and Hubert) were applied to the chronosequences of the annual P and Q over the studied period (1951–1995) using Khronostat 1.01 software in order to characterize the steadiness periods. The rank test describes the hazardous character of the chronological series per value of the calculation variable. Break tests (done with Buishand, Pettitt, and Lee-Heghinian or Bayesian method) allowed a date to be obtained due to the critical value of the Pettitt test [70] or the break point of the Bayesian method [71]. The Hubert segmentation then gave the year of the detected breaks over the series, with the average value and the standard deviation of different segments [72]. This segmentation also helped to identify some periods of stasis [26,38,43,72,73]. These homogeneous rainfall and discharge periods are related to homogeneous hydroclimatic periods for the studied basin. After this, some linear and polynomial regressions were adapted, respectively, to P and Q, with the aim of evaluating the degree of correlation between them using the Student's test and the Pearson method.

Depletion coefficients and the total potential groundwater discharge were calculated with the daily Q [44,65] to follow the spatiotemporal dynamics of water resources. The mathematical expression of the river depletion coefficient is written as follows:

$$Q_t = Q_0 e^{-kt}$$

where $Q_t$ = discharge at time t (in our study, the day), $Q_0$ = initial discharge (discharge at the beginning of the depletion), and k = Maillet depletion coefficient. Maillet's law is modeled on a single reservoir contributing to flow in the watershed under study [26,44]. The integration of this equation over the interval (1–365) [26] gives an estimate of the volume mobilized from the sub-basins and basin aquifers:

$$V mobilized = \int_0^{+\infty} Q_0 e^{-kt} dt = \frac{Q_0}{k}$$

This approach reconsiders an annual dynamics state during the time of groundwater contribution to support the recorded flows and water resources level per targeted outlet to determine their fragility degree.

## 3. Results

### 3.1. Temporal Evolution and Breaks in Annual Rainfall

#### 3.1.1. Annual Rainfall Evolution

The interannual mean rainfall of the three studied sub-basins in the Ouham Basin was relatively homogeneous whatever the considered hydroclimatic periods. We found a weak decrease in rainfall between the plateau zone and the plain downstream, expressing the following weak altitude effect: 1423 mm at Bozoum, 1439 mm at Bossangoa, and 1393 mm at Batangafo, respectively. Indeed, in the region, the rainfall drought reduced the isohyets according to the North–South gradient [74]. The Ouham basin is oriented West–East (Figure 1), and the West–East evolution of rainfall from upstream to downstream be only due to the elevation factor. The annual rainfall maximums, respectively, were 1807 mm (1955–1956) in the sub-basin of Bozoum, 1715 mm (1955–1956) in that of Bossangoa, and 1684 mm (1993–1994) in the whole Ouham River Basin at Batangafo. In the upper and middle sub-basins, these annual maximums were recorded in the wet period even though downstream, at Batangafo, this one was observed during the dry period. However, the minimums were all recorded in the same year for the three studied sub-basins, in 1984–1985: 1027 mm at Bozoum, 1130 mm at Bossangoa, and 1102 mm at Batangafo, i.e., during the drought accentuation in the 1980s [46].

The rainfall index showed a declining linear trend (R = 0.46 at Bozoum, 0.49 at Bossangoa, and 0.46 at Batangafo) due to evidence of three common rainfall periods marked in each of the sub-basins: 1951–1971, 1972–1990, and 1991–1995 (Figure 2). The 1951–1971 period was wet. However, the 1972–1990s was driest, showing −8% deficit (Ouham at Bozoum) and −7% (Ouham at Bossangoa and at Batangafo) deficits compared to the interannual means per sub-basin. It is noticeable that the late short period (1991–1995) was rainy, with an average recording of 1462 mm (i.e., +5%) in the whole basin. Some increasing exceedances from upstream to downstream per sub-basin were also recorded: 23 mm (+2%) at Bozoum, 59 mm (+4%) at Bossangoa, and 69 mm (+5%) at Batangafo. These figures confirm the spatial homogeneity of the rainfall in the whole Ouham basin.

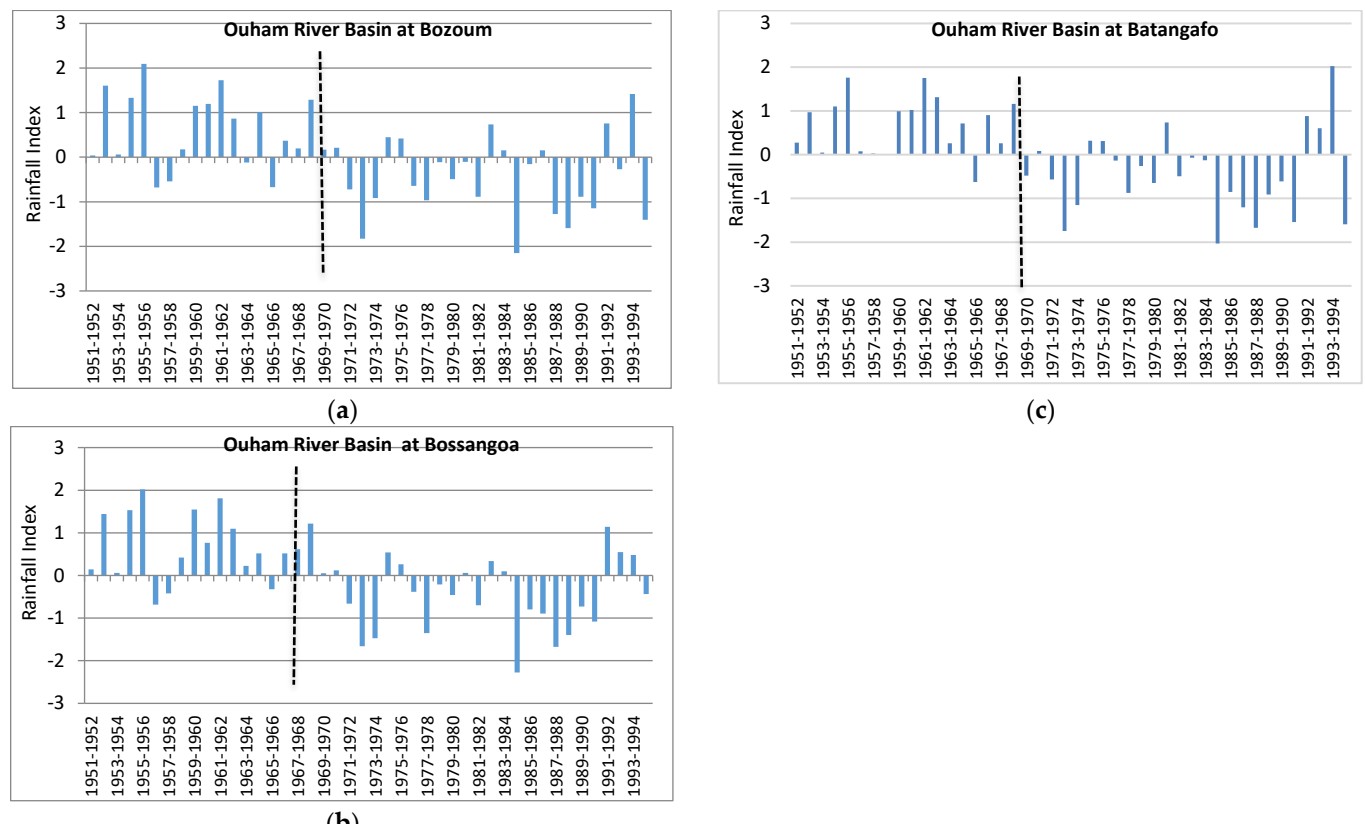

**Figure 2.** Rainfall index and break point in the rainfall series of the Ouham River Basin at (**a**) Bozoum, (**b**) Bossangoa, (**c**) and Batangafo.

This rainfall evolution between the three sequences (1951–1971, 1972–1990, and 1991–1995), obtained via the rainfall indices from the sub-basins of Ouham at Bozoum and at Bossangoa, and even at Batangafo, was confirmed by the averaged SPI in the Ouham Basin (Figure 3). The mean SPI globally shows that there was a moderately wet trend after 1951, and another moderately dry trend from the start of 1971 to 1990 (Bozoum and Batangafo) or 1972 (Bossangoa) at Bozoum (Figure 3a), at Bossangoa (Figure 3b) and at Batangafo (Figure 3c), respectively, displaying some drought annual peaks (indices > −1.5). These results do underline that hydrological drought was gradually accentuated over the 1971/1972–1995 period. The results are almost the same as those obtained using the rainfall index once the difference between these trends is precisely related to the gradual severity in drought occurring mainly in the 1970s and 1980s. Also they do confirm the rainfall recovery of the 1990s starts in the whole catchment. These results reveal that the drought effect on the overall rainfall was moderate. Between both episodes (wet and dry), each trend duration was almost the same, although drought was amplified in the 1980s (Figure 3). During the wet period, some moderate to normal drought was observed in the 1956–1958

period in the sub-basins at Bozoum and at Bossangoa, and the whole basin at Batangafo. Conversely, in the dry period, drought peaks were also noticed in the 1970s, as in the 1980s. In the 1970s, the SPI showed a moderate drought mainly in 1973 at Bozoum (−1.71) and at Batangafo (−1.61), with mild drought in 1977–1981 and 1976–1979, respectively; the Bossangoa sub-catchment also recorded mild drought from 1976 to 1981. In the 1980s, moderate drought was observed, respectively, at Bozoum (1984–1990) and at Batangafo (1983–1990); nevertheless, drought peaks and severe droughts were also recorded, such as in 1987 (−1.67) at Bossangoa, and in 1985 (−1.32), 1987 (−1.72) and 1994 (−1.64) at Batangafo. The temporal behavior was found to be quite similar for the three sub-basins with annual drought peaks in 1973, 1985, and 1987.

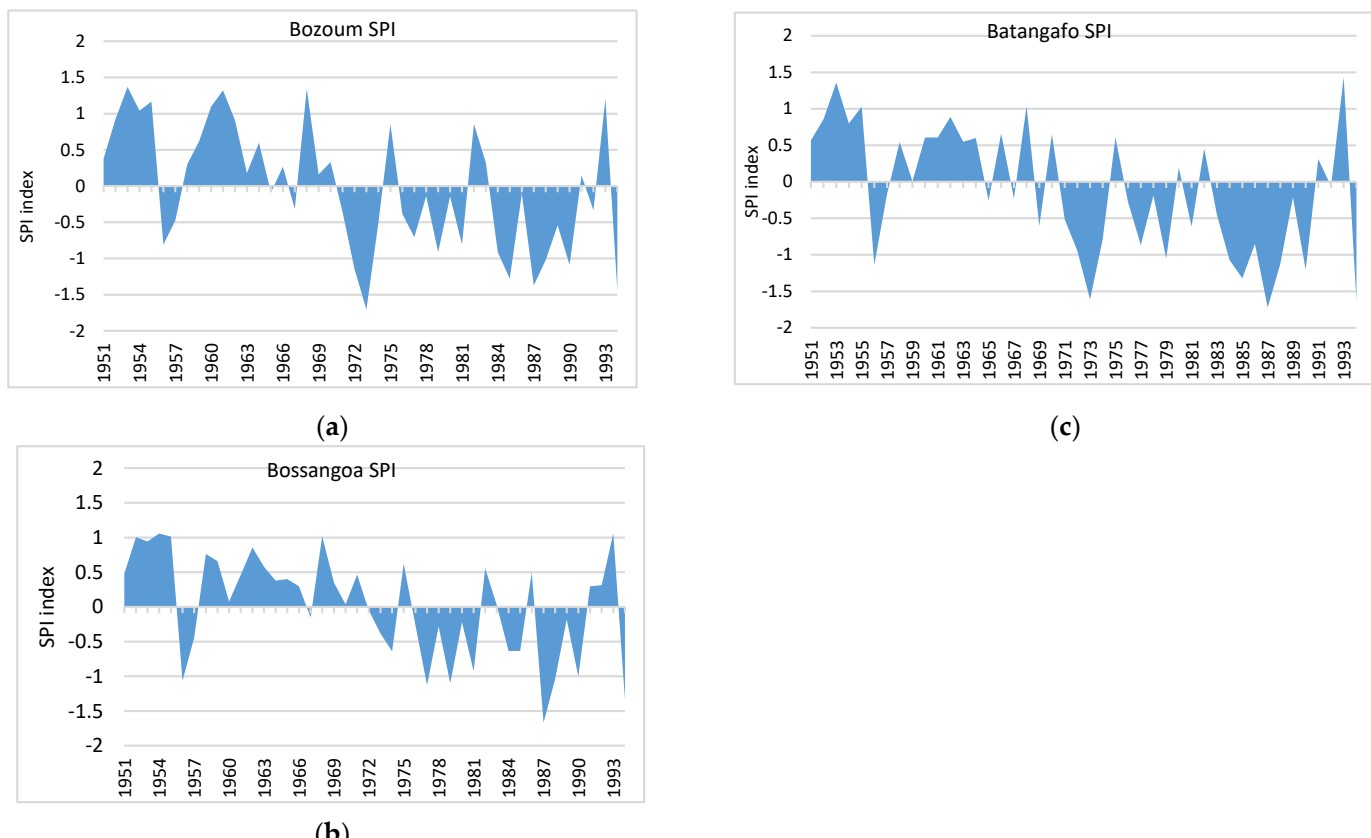

**Figure 3.** SPI index over the Ouham River Basin at (**a**) Bozoum, (**b**) Bossangoa, and (**c**) Batangafo.

The interannual monthly average over the 1951–1995 period was found to be quite similar for the three stations: 119 mm at Bozoum, 120 mm at Bossangoa, and 116 mm at Batangafo (Figure 4). Thus, the calculated annual and monthly rainfall regimes between the 1951–1970 and 1971–1995 periods, i.e., around the 1970 climatic break, have only varied by less than 3 mm (Figure 4a–c). These values were 16 mm at Bozoum, 13 mm at Bossangoa, and 13 mm at Batangafo, giving a mean decline of 14 mm in all basins. The declining monthly rainfall heights in both periods (1951–1970 and 1971–1995) were −13% in the Bozoum sub-basin, −10% in the Bossangoa sub-basin, and −11% in the whole basin at Batangafo. They imply a time lag of one month at the start of the rainy season, from April to May, and a reduction in its months, from 7 (April–October) to 6 (May–October). These three annual rainfall periods were confronted with homogeneous periods in these series in order to evaluate their validity. Otherwise, the spatiotemporal evolution of the rainfall series over the Ouham River basin between wet and dry periods subjected them to both breaks and homogeneous averages, which are possible to detect per sub-basin.

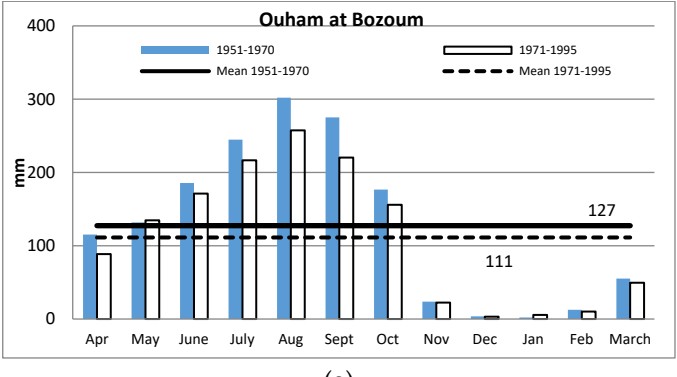

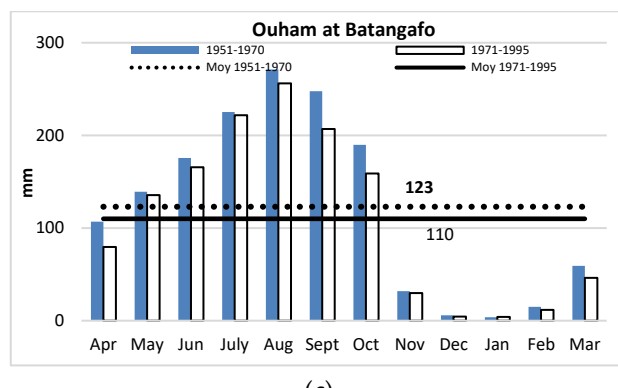

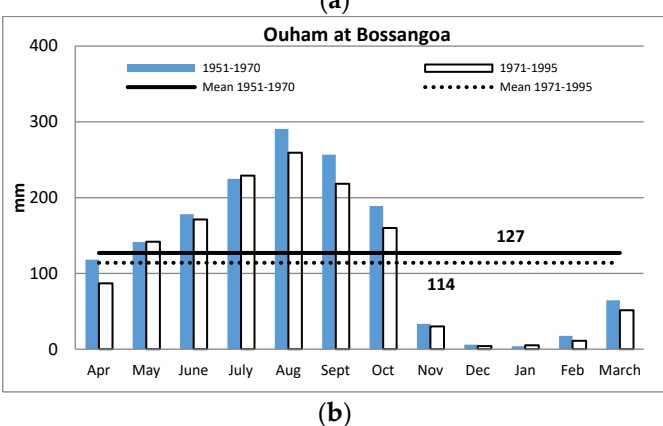

**Figure 4.** Monthly average rainfall regime before and after 1970–1971 in the Ouham River sub-basins at (**a**) Bozoum and at (**b**) Bossangoa, and the Ouham basin at Batangafo (**c**).

### 3.1.2. Breaks and Homogeneous Rainfall Periods

The rank test establishes the hazardous character of the studied series (rejected null hypothesis at a confidence threshold of 99%, 95%, and 90%) for the three stations. Buishand's test did not identify any break in these series. While the Pettitt test described one in 1970 (at Bozoum and Bossangoa) and another in 1968 at Batangafo, The Lee–Heghinian test confirms the break dates of 1970 (at Bozoum) and 1968 (at Batangafo and Bossangoa) (Table 2). Also, the Hubert segmentation detected a break in 1969, and two homogeneously averaged rainfall in the Bozoum sub-basin (1534 mm and 1336 mm), and in 1967, homogenously averaged rainfall in both Bossangoa (1538 mm and 1374 mm in the 1951–1967 and 1968–1993 periods respectively) and Batangafo (1493 mm and 1327 mm also in the 1951–1967 and 1968–1993 periods respectively). The observed break at Bossangoa and at Batangafo was earlier (1967), compared to the occurrence date at Bozoum in 1969. Thus, the break test allows for the identification of two homogenous periods for each rainfall series, namely, wet and dry. The wet period, before the 1970s, is marked by an exceedance value of +8% at Bozoum, and +7% at Bossangoa and Batangafo, compared to the interannual mean. The dry period is marked by the weakest reductions, namely −6%, −4% and −5%, respectively.

In conclusion, a rainfall break in the 1951–1995 period could be confirmed around 1970 for the three studied stations. In the Ubangi Basin, a rainfall break was also identified in 1970, with a mean deficit of −5% [21,25,26].

**Table 2.** Statistical segmentation of annual rainfall series (1951–1995).

| Period | Interannual Mean (mm) | Pettitt [70] | Lee and Heghinian [71] | Hubert Segmentation [72] | Standard Deviation | Observed Period | Report to Interannual Mean (%) |
|---|---|---|---|---|---|---|---|
| Ouham at Bozoum (8100 km$^2$) | 1423 | 1970 | 1970 | 1951–1969: 1534 | 153 | Wet | +8 |
| | | | | 1970–1993: 1336 | 163 | Dry | −6 |
| Ouham at Bossangoa (22,800 km$^2$) | 1439 | 1970 | 1968 | 1951–1967: 1538 | 110 | Wet | +7 |
| | | | | 1968–1993: 1374 | 114 | Dry | −4 |
| Ouham at Batangafo (43,650 km$^2$) | 1393 | 1968 | 1968 | 1951–1967: 1493 | 96 | Wet | +7 |
| | | | | 1968–1993: 1327 | 135 | Dry | −5 |

*3.2. Temporal Evolution and Breaks in the Annual Flow*

3.2.1. Dynamics of the Annual Mean Q of the Ouham River per Outlet

The interannual mean discharges of the Ouham were 78 m$^3$/s at Bozoum (8100 km$^2$), 192 m$^3$/s at Bossangoa (22,800 km$^2$) and 282 m$^3$/s at Batangafo (43,650 km$^2$) over the 1951–1995 period (Figure 5). These correspond to specific discharges of 10 l/s/km$^2$, 8 l/s/km$^2$, and 7 l/s/km$^2$, respectively, which represent the natural evolution of the river regime along its flow channel. Therefore, ecological continuity is noticed in the spatial functioning of the Ouham River. The maximums of the annual mean discharges are recorded during the wet period of 1951–1969, only for Bozoum and Bossangoa: in 1955–1956 at Bozoum (141 m$^3$/s), in 1962–1963 at Bossangoa (372 m$^3$/s), and in 1963–1964 at Batangafo (669 m$^3$/s). The minimums are recorded during the dry period of 1970–1994: in 1979–1980 at Bozoum (20 m$^3$/s) and at Bossangoa (55 m$^3$/s), and during the wet period at Batangafo in 1992–1993 (26 m$^3$/s).

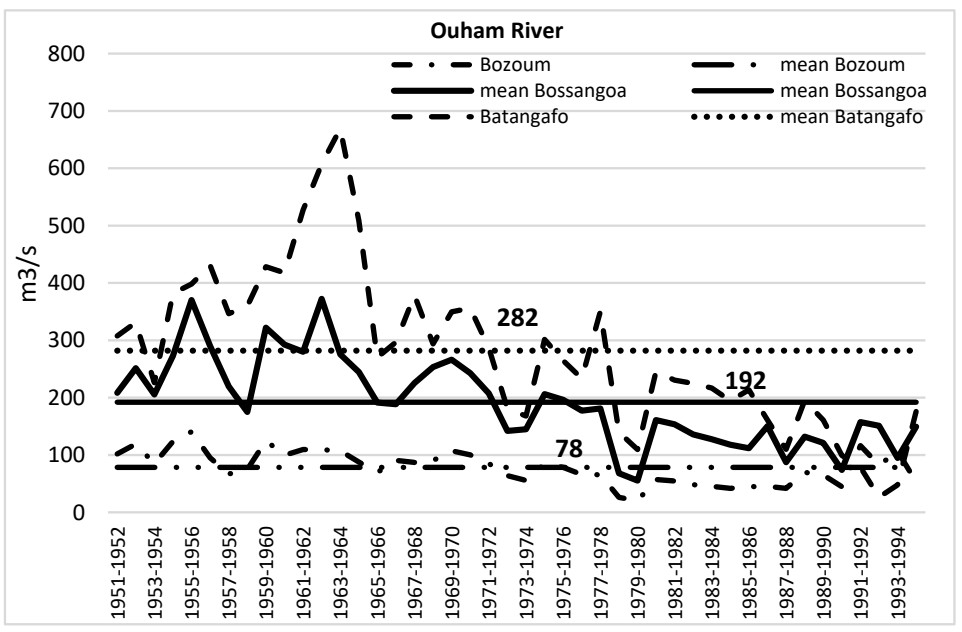

**Figure 5.** Interannual evolution of the annual discharges of the Ouham River at Bozoum, Bossangoa, and Batangafo stations.

In detail, the calculated flow index (Figure 6) allows us to describe three homogeneous hydrologic periods for the three studied outlets. The evolution of the flow index is similar in the three stations, with the only difference being from 1990 at Bozoum in the upper part of the Ouham River (Figure 6). The flow index evolution shows three periods: a period of very positive excess of approximately 1 from 1951 to 1964; a mean period of approximately 0 from 1965 to 1978; and the driest period around an index of −1 from 1979 to 1990 at Bozoum and to 1994 at the two downstream stations. It seems that the variations are more important from upstream to downstream, and that they describe a relatively constant decrease in annual discharges from 1951 to 1994, with a deficit break in 1979, such as that which occurred for the rainfall data series. Therefore, the temporal evolution of Q per outlet displayed a decreasing trend, except in Bozoum from 1990. The linear regression was good in Bossangoa (R = 0.77), pretty good in Batangafo (R = 0.68), and tolerable in Bozoum (R = 0.52).

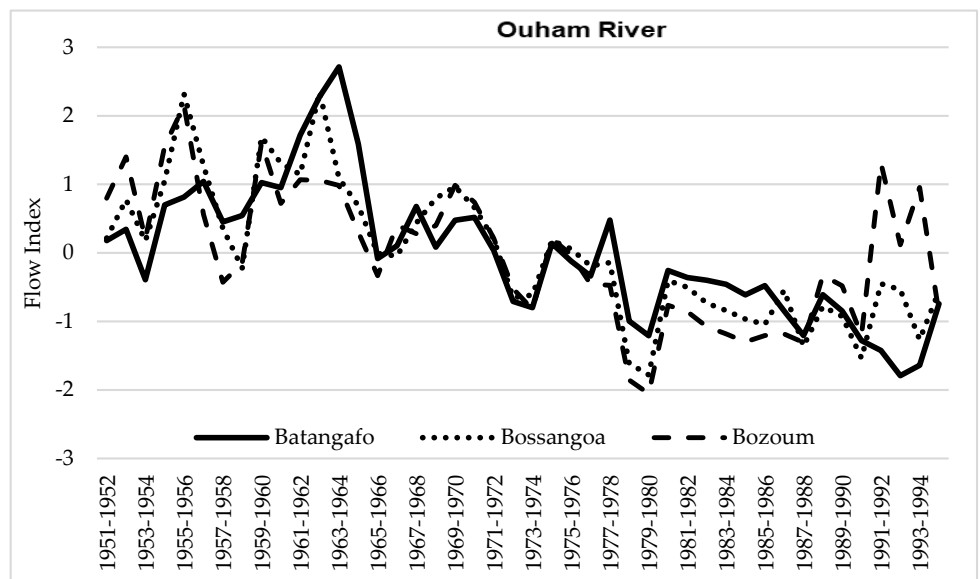

**Figure 6.** Flow index of the Ouham River at Bozoum, Bossangoa, and Batangafo.

In the monthly scale (Figure 7), the interannual mean discharges were 78 m$^3$/s in Bozoum, 192 m$^3$/s in Bossangoa, and 280 m$^3$/s in Batangafo, increasing with the sub-basins or basin size. The calculated annual and monthly discharges between the 1951–1970 and 1971–1995 periods (wet and dry, respectively) were also significant in terms of the water volume from upstream to downstream (Figure 7a–c): the decreasing mean intervals per outlet, respectively, were 39 m$^3$/s in Bozoum, 119 m$^3$/s in Bossangoa and 210 m$^3$/s in Batangafo, making 5 l/s/km$^2$ per outlet. This is translated into a monthly Q decline per outlet of −39% at Bozoum, −46% at Bossangoa, and −53% at Batangafo, respectively, between 1951–1970 and 1971–1995; this shows an accentuated gradual reduction from upstream to downstream for both periods. Therefore, the Q reduction rates are three to four, even five, times higher than the monthly rainfall per sub-basin. This demonstrates the complexity of the phenomenon of the reduction in discharge compared to rainfall in the entire catchment. This reduction in the monthly mean discharges in both periods reveals the late start of the rising season, being in July rather than in June, and the reduction in the high-level water period, being from July to November (5 months), instead of June to November, in the 1951–1970 period. What do these reductions imply in terms of ruptures in these annual mean discharges?

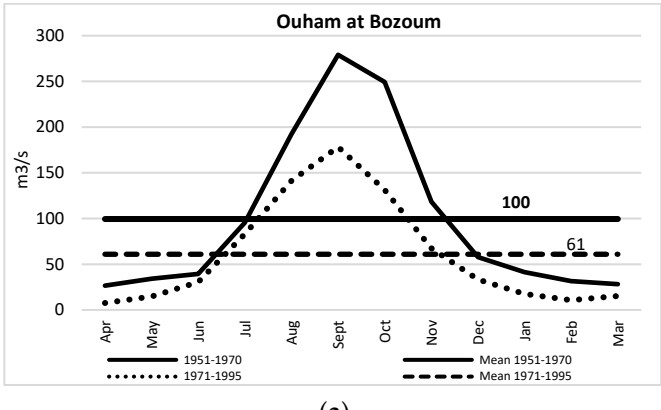

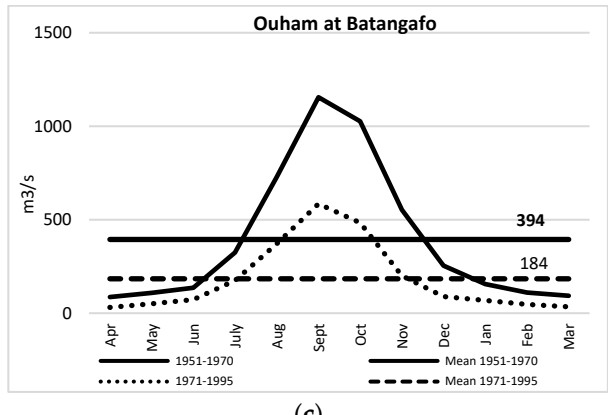

(**a**)

(**c**)

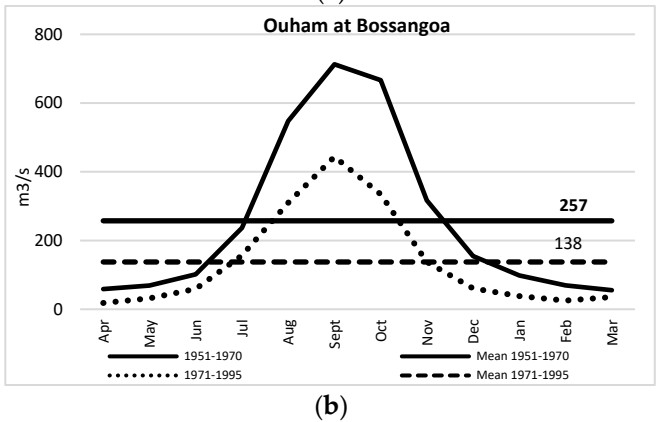

(**b**)

**Figure 7.** Monthly average flow regime before and after 1970–1971 for the Ouham River at (**a**) Bozoum, (**b**) Bossangoa, and (**c**) Batangafo.

### 3.2.2. Breaks and Homogeneous Hydrological Periods

The rank test establishes the hazardous character of the Q series (rejected null hypothesis at confidence thresholds of 99%, 95%, and 90%) for the three stations. The Buishand test did not show any rupture. Meanwhile, both the Pettitt and Lee–Heghinian tests found a common date in the three outlets of 1971; this was except for Batangafo, where the break was noted to be in 1970 (Table 3). In return, the Hubert Segmentation determined two dates in Bozoum (1970 and 1989) and Bossangoa (1963 and 1976), and four dates in Batangafo (1959, 1963, 1970 and 1988) (Table 3). Except for 1970 (in Bozoum and Batangafo), the Hubert segmentation did not confirm the dates given by the Pettitt and Lee–Heghinian tests. Thus, the Hubert segmentation detected homogeneous annual mean discharges and periods per outlet, between wetness and/or drought: there were three in Bozoum and Bossangoa, and five in Batangafo (Table 3). In detail, the homogeneous periods corresponding to hydrological abundance were wet to very wet in the 1951-1970 period at Bozoum and Batangafo respectively, and from 1951 to 1976 at Bossangoa. After this wet period, there was a dry period everywhere until 1989. Beyond 1989, except in Bozoum, a hydrological exceedance was observed in 1990–1992 (+14%), compared to the interannual average; however, this period was very dry for the two other stations. Therefore, we first keep in mind that the regional climatic rupture of 1970 was noted in the whole Ouham River Basin, and second, that there was a divergence in the discharge evolution between the upper part at Bozoum, which was very wet from 1990, and the two downstream stations of Bossangoa and Batangafo, which were very dry from this date.

**Table 3.** Statistical segmentation of annual Q series (1951–1995).

| Period | Interannual Mean (m³/s) | Pettitt [70] | Lee and Heghinian [71] | Hubert Segmentation [72] | Standard Deviation | Observed Period | Report to Interannual Mean (%) |
|---|---|---|---|---|---|---|---|
| Ouham at Bozoum | 78 | 1971 | 1971 | 1951–1970: 99 | 19 | Wet | +27 |
| | | | | 1971–1989: 54 | 17 | Dry | −31 |
| | | | | 1990–1992: 101 | 28 | Wet | +14 |
| Ouham at Bossangoa | 192 | 1971 | 1971 | 1951–1963: 275 | 58 | Very wet | +43 |
| | | | | 1964–1976: 202 | 38 | Wet | +5 |
| | | | | 1977–1993: 120 | 34 | Very dry | −37 |
| Ouham at Batangafo | 282 | 1971 | 1970 | 1951–1959: 369 | 65 | Wet | +31 |
| | | | | 1960–1963: 578 | 74 | Very wet | +105 |
| | | | | 1964–1970: 319 | 42 | Wet | +13 |
| | | | | 1971–1988: 206 | 62 | Dry | −27 |
| | | | | 1989–1993: 85 | 57 | Very dry | −70 |

These results clearly show a gradual hydrological reduction, after 1951, despite a "hydrological bound" after 1963 in Bossangoa (1964–1976) and Batangafo (1964–1970). The rupture curve shows varied trends: the highest R values are in Bossangoa (0.94) and Batangafo (0.83), rather than in Bozoum (0.55), which establishes in time this progressive and severe (mainly in Batangafo) hydrological decline.

Therefore, this study reveals a disparity between the R and Q relations, with higher R upstream (0.68 in Bozoum and 0.66 at Bossangoa) than downstream (0.42 in Batangafo), caused by the effect of spatial heterogeneity and the high level of evaporation in this Sudanian area. This means that the flow regime is not subjected to rainfall only, but also to basin permeability, soil land use/change, etc. How, then, is the diverse response of the studied variables reflected in the basin aquifers?

*3.3. Depletion Coefficients and Mobilized Volume of Aquifer*

The averaged depletion coefficients are 0.022 day$^{-1}$ at Bozoum, 0.024 day$^{-1}$ at Bossangoa, and 0.020 day$^{-1}$ at Batangafo over the studied period. Before 1951–1970 and after 1970 (1971–1995), these values were, respectively, 0.018 day$^{-1}$ and 0.027 day$^{-1}$ at Bozoum (Figure 8a), 0.019 day$^{-1}$ and 0.029 day$^{-1}$ at Bossangoa (Figure 8b), and 0.019 day$^{-1}$ and 0.022 day$^{-1}$ at Batangafo (Figure 8c). Their extreme values are 0.041 day$^{-1}$ (1992–1993) and 0.0147 day$^{-1}$ (1963–1964) at Bozoum, 0.0575 day$^{-1}$ (1989–1990) and 0.0127 day$^{-1}$ (1978–1979) at Bossangoa, and 0.0338 day$^{-1}$ (1988–1989) and 0.0045 jour$^{-1}$ (1992–1993) at Batangafo. The highest coefficients of aquifer depletion always appear in the 1986–1995 period, and earlier in the downstream area and later in the upstream area. This means that the impact of drought on the river flow appeared first at Batangafo in the downstream part of the basin, then at Bossangoa, and at last at Bozoum, in the upstream part. To compare with the Ubangi River Basin, the maximum aquifer depletion was observed in 2000–2001 (0.025 day$^{-1}$) [26]. At this time, the Ouham River Basin seems to have suffered the drought much more than the Ubangi River Basin. Consequently, the basin water resources declined in the whole Ouham River Basin, from approximately 1965 at Bozoum, Bossangoa, and Batangafo, a few years before the hydrological break of 1970. In order to obtain an image of the actual volume of the groundwater reserve in each studied basin, we calculated the water volume in the period 1986–1992, and the following values were obtained: 0.5 km³, 0.94 km³, and 1.6 km³, respectively (Table 4), for average reserves of 1.5 km³, 3.2 km³, and 5.2 km³; these were calculated during the period 1951–1970, corresponding to the wet period. This means that there has been a very significant decrease in the aquifers since 1970, namely, a decrease of three times the basin water resources.

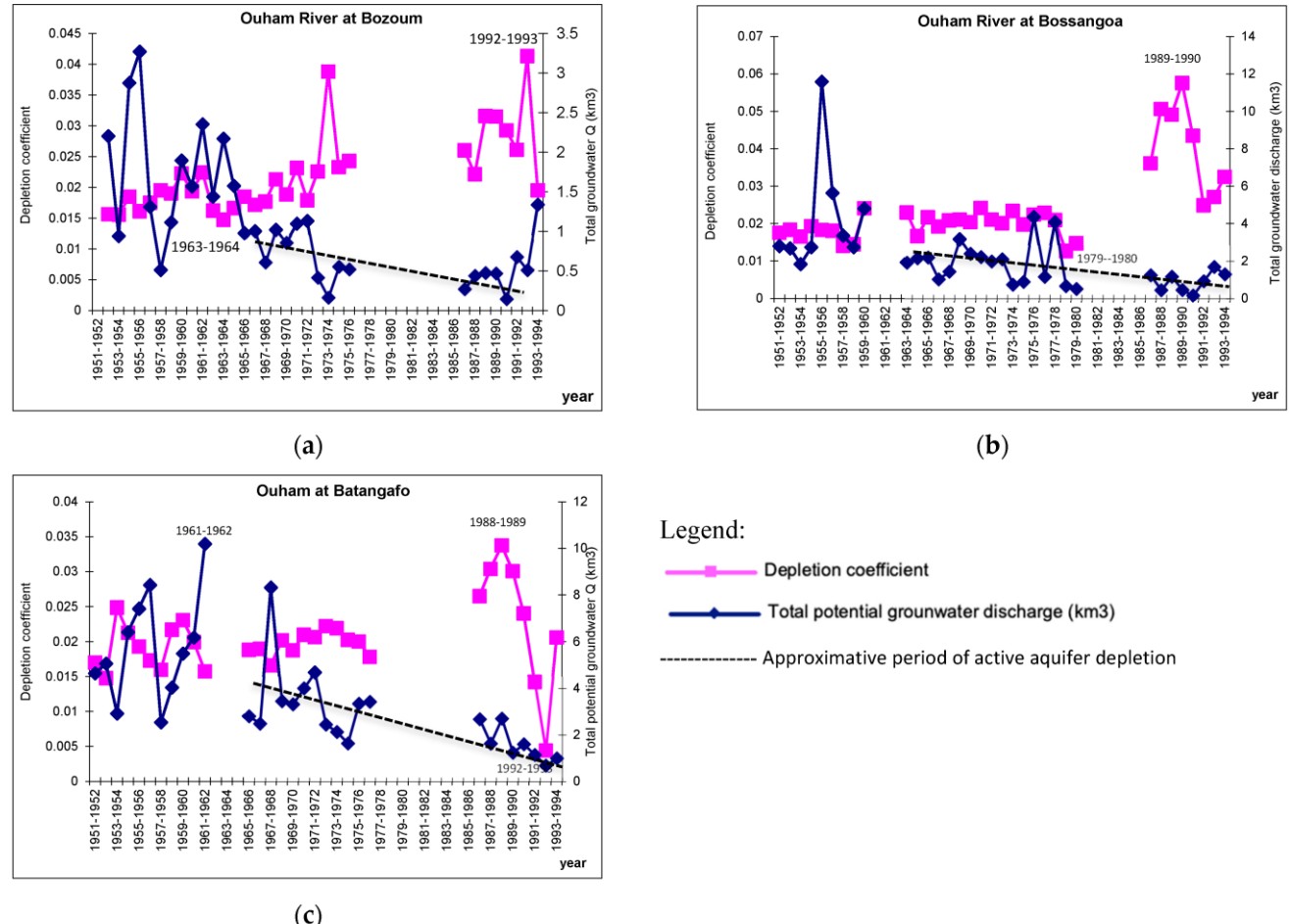

**Figure 8.** Evolution of depletion coefficient and total potential groundwater of the Ouham River Basin at (**a**) Bozoum, (**b**) Bossangoa, and (**c**) Batangafo over the 1951–1995 period.

**Table 4.** Evolution from upstream to downstream of water resources into sub-basins per period and the Ouham basin (1951–1995).

| Period/Basin | Ouham at Bozoum (km$^3$) | Ouham at Bossangoa (km$^3$) | Ouham at Batangafo (km$^3$) |
|---|---|---|---|
| 1951–1970 | 1.5 | 3.2 | 5.2 |
| 1986–1995 | 0.5 | 0.94 | 1.6 |

## 4. Discussions

### 4.1. Annual Rainfall and Discharge Relationship in the Ouham River Basin

For a slight shift in the rainfall regime, the Ouham River flow decreased at its main outlet (Batangafo), from 440 m$^3$/s on average before 1969–1970 to less than 250 m$^3$/s in the following years [30], meaning that there was an approximately −43% reduction in the initial annual discharge due to drought. This emphasizes that a weak reduction in rainfall (−5%) leads to high hydrological deficits (from −30% to −43%), underlining a complex P–Q relationship. This is characterized by weak regression coefficient R values (Table 5): 0.42 at Batangafo, 0.68 at Bozoum, and 0.66 at Bossangoa. As is well known in many other hydrological basins in the world, these values highlight that, in the Ouham River Basin, other factors (mainly vegetal cover and land use) interact with the Ouham hydrology. The P–Q correlation decreased after the drought break of 1970, especially at Batangafo, where varying annual rainfall volume provided a large range (Figure 9). The graphical analysis showed that the Q value distribution is relative to the wet and dry hydroclimatic

period considered: 1970–1971 (1150 mm for 356 m³/s); 1988–1989 (1119 mm for 195 m³/s); 1989–1990 (1207 mm for 161 m³/s) and 1992–1993 (1667 mm for 26 m³/s).

**Table 5.** P–Q correlations per sub-period and sub-basin in the Ouham River Basin (1951–1995).

| Sub-Basin/Period | 1951–1995 | 1951–1970 | 1971–1995 | Trends |
|---|---|---|---|---|
| Bozoum | 0.68 | 0.71 | 0.40 | Linear |
| | 0.71 | 0.72 | 0.52 | Polynomial |
| Bossangoa | 0.65 | 0.55 | 0.23 | Linear |
| | 0.68 | 0.64 | 0.23 | Polynomial |
| Batangafo | 0.44 | 0.35 | 0.15 | Linear |
| | 0.46 | 0.36 | 0.48 | Polynomial |

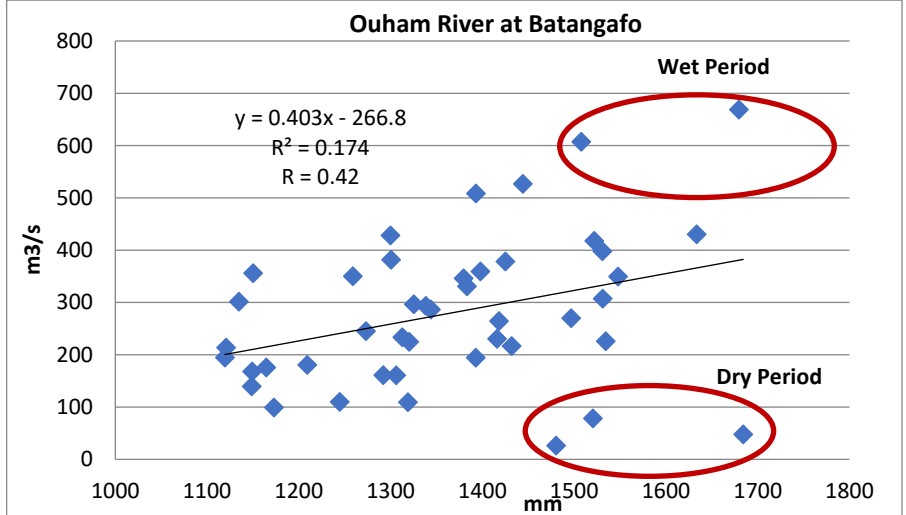

**Figure 9.** Disparities in the rainfall–runoff relationship in the Ouham River Basin at Batangafo.

The increase in rainfall from 1991 (Table 3) led to an increase in the river discharge only at Bozoum and Bossangoa, and not in the downstream part of the Ouham River Basin. This is explained by the large, irrigated areas built by the end of the 80s around Batangafo [30]. Thus, some R–Q correlations were calculated per outlet, with linear and polynomial trends in the 1951–1995 period and its sub-periods, distributed between wet (1951–1970) and dry (1971–1995) periods (Table 5). Globally, the R coefficients were high in Bozoum (R = 0.68 and 0.71), mainly in the 1951–1970 period (R = 0.71 and 0.72), and were degraded from upstream (Bozoum) to downstream (Batangafo). In the 1951–1995 period, Bozoum and Bossangoa had high R coefficients compared to Batangafo, even in the wet period (1951–1970). If the weak R characterizes the dry period (1971–1995), the polynomial trend is higher than the linear one. This would be the noticeable annual variability effect in the R and Q studied series.

These hydrological deficits also mark the upper Chari River. Indeed, some rainfall deficits of −8% and −3% also induced severe hydrological shortfalls of −39% on the Gribingui at Kaga-Bandoro, a sub-tributary of the Chari via the Bamingui River, as on the Fafa at Bouca [46,75]. This implies a noticeable reduction in the mobilized volume of the aquifer per sub-basin. Such regressive hydrological dynamics in the Ouham mainly increased in the 1980s and 1990s, proof of the unsteadiness of the regime, between high deficit (−70%) and water abundance. Over this basin area of 43,650 km², rainfall is relatively homogeneous, without a relief effect. Nevertheless, the normal evolution of the hydrological regime through the specific discharges was marked by an unhooking effect between the three stations from one to another in 1985–1986 (Figure 10). Indeed, the flow

increased at Bozoum, while it was more or less stable at Bossangoa, and both declined and rose at Batangafo. Is this phenomenon due to peoples' increasing impact on these outlets regarding wetness/drought?

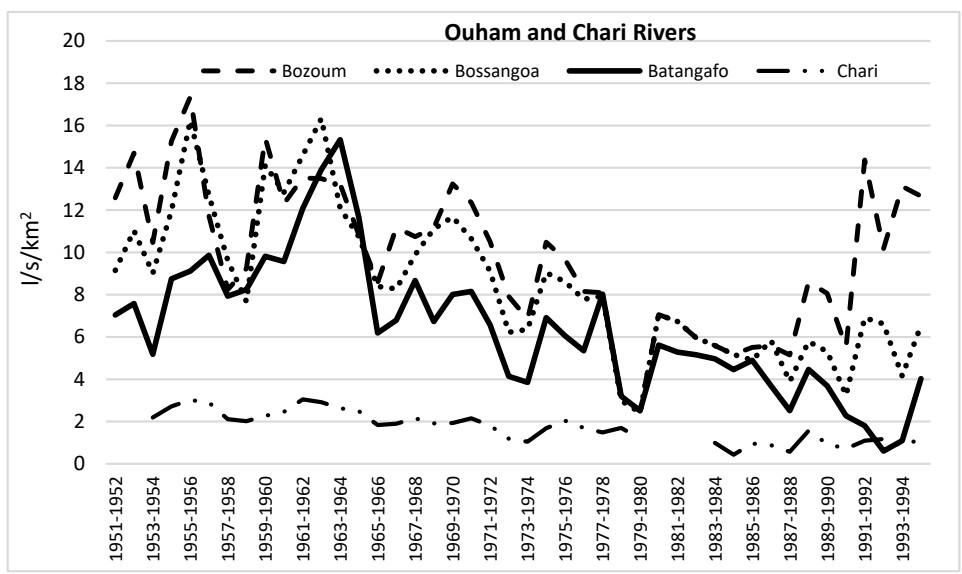

**Figure 10.** Specific flow into outlets of the Ouham River and Chari River at Ndjamena.

*4.2. Breaks Comparison between Ouham and Ubangi Rivers*

The breaks detected using Hubert segmentation helped to distinguish homogeneous hydroclimatic periods (wet, normal, and dry). The segmentation was used to confirm the breakpoints and exceedance probabilities of the test's critical values, as well as the annual rainfall and annual discharge.

In the Ubangi River Basin, a rainfall break was found in 1971: wet in 1935–1970 and dry in 1971–2015 [26]. This is rather similar to the sub-basins of the Ouham at Bozoum (1969), Bossangoa (1967), and Batangafo (1967). Although different, these dates present a small time lag from one to two years. They are earlier in the Ouham River Basin in the north (1967 and 1969 with Hubert segmentation) compared to those of the Ubangi River Basin at Bangui (1935–2015) in the South. Nevertheless, the rainfall deficits are quite the same in both basins, and are relatively weak (−5%) in comparison to other regions in West Africa [23,39]. As for annual P, the Q break dates over Ubangi are quite similar to those over Ouham but marked diversely due to the respective location of their hydrometric station. Indeed, two opposed hydrological sequences (dry and wet) are recorded in the Ouham River Basin at Bozoum (dry in 1971–1989 and wet in 1990–1992). On the other hand, two dry sequences are recorded successively (1971–1988 and 1989–1993) in the Ouham River Basin at Batangafo, similar to the Ubangi River Basin: 1971–1982 and 1983–2013 [26]. These show severe hydrological deficits in both cases. Thus, we can mention that the hydroclimatic periods of the Ubangi River Basin, such as in the Congo River at Brazzaville [76], correspond to those recorded in the Ouham River Basin at Batangafo in the upper basin of Lake Chad.

The fact that the Ubangi River Basin at Bangui is a twelfth wider than the Ouham River Basin at Batangafo does impact the segmentation of the hydroclimatic periods.

*4.3. Water Resources Statement and Effect of Ongoing Drought*

4.3.1. Depletion and Groundwater Resources in the Region

The depletion coefficients are higher over the Ouham (during 1986–1995: 0.041 day$^{-1}$ at Bozoum, 0.058 day$^{-1}$ at Bossangoa, and 0.034 day$^{-1}$ and 0.0045 day$^{-1}$ at Batangafo) than on some rivers in the upper basin of the Chari in the CAR (0.017 day$^{-1}$ on the Gribingui at Kaga-Bandoro (5680 km$^2$), and 0.022 day$^{-1}$ on the Fafa at Bouca (4380 km$^2$), another tributary of the Ouham [46,75]. The coupled evolution of the depletion coefficient and

aquifer volume (Figure 8) signals three facts globally. The first is marked by an alternated fluctuation in the two variables in 1960 (Ouham at Bossangoa), 1965 (Ouham at Bozoum), and 1968 (Ouham at Batangafo), their unhooking dates, and the relative abundance of mobilized volume per aquifer. The second fact expresses the unhooking of variables, mainly the depletion accentuation in all outlets between 1963 and 1964 (Bozoum and Bossangoa) and 1968 and 1969 (Batangafo). The last characterizes an exacerbation of the depletion phenomenon, while the mobilized volume of the aquifer drastically drops due to the effect of drought. These results expose the state of certain water resources in this fluvial system.

### 4.3.2. Water Resources' Evolution per Outlet and Recent Trends

To assess the current effect of drought on water resources, we calculated the specific discharges per sub-basin, which confirmed an upstream–downstream degradation of water resources, paradoxically to sub-basin size, with the following interannual specific discharge $Q_s$: 11 $l \cdot s^{-1} \cdot km^{-2}$ at Bozoum, 8.48 $l \cdot s^{-1} \cdot km^{-2}$ at Bossangoa and only 6.5 $l \cdot s^{-1} \cdot km^{-2}$ at Batangafo. Identical dynamics are observed on the Chari River, with 1.74 $l \cdot s^{-1} \cdot km^{-2}$ at Ndjamena (Figure 10). The specific discharge $Q_s$ observed for Chari is four times weaker than that observed for the Ouham River at Batangafo. These results show an inverse situation based on water volume decline via the Qs reduction in the Chari River Basin upstream to downstream, rather than their increase according to the sub-basins' size. The $Q_s$ multiannual evolution underlines a high global trend regarding the interannual mean values until 1971–1972 for Ouham at Bozoum, until 1975–1976 for Ouham at Bossangoa, until 1977–1978 for Ouham at Batangafo, and until 1983–1984 for Chari at N'Djamena, which dropped and increased, despite being high over Ouham at Bozoum from 1992 to 1995.

In Bozoum and Bossangoa, the maximum and minimum $Q_s$ are high compared to those ones and dates' occurrence in the Batangafo outlet. Thus, all water resources worsening are noticed from Bozoum to Batangafo, via Bossangoa i.e., from West to East into the whole basin Is this phenomenon dissociated from the ongoing declining context or does it purely belong to the natural functioning of this basin? The weakest $Q_s$ of all studied in this series exposed the effect of hydrological drought or very dry years. Nevertheless, the observed rainfall reprise in the basin had a positive impact on the $Q_s$ at Batangafo and also at N'Djamena (Figure 10).

### 4.4. Impact of Ouham Water Resources on Lake Chad

These trends are also observed in the evolution of annual discharge in the Chari/Logone hydrological system at N'Djamena [28], showing the high runoff peak in 1961 (Figure 11). Indeed, 1961 was the wettest year and it saw floods in the whole CAR [77,78], even in Africa [79]. However, both 1973 and 1984 saw some severe levels of drought in the LCB, reaching a peak in 1984 over the Chari/Logone and the Ouham. Indeed since 1971, the diverse spatiotemporal impact of this ongoing drought was recorded/established on both the Ouham and Chari rivers in the whole LCB (Figure 6).

This trend was also positive for the level of Lake Chad, because [28] noticed that the southern areas of the basin, possess an important flow. In this way, the decrease in discharge during the 1980s reflects a reduction in the lake surface, despite the fact that rainfall declined weakly. This allowed [18] to note that, since 1995, Lake Chad has seen a steady return of water in the northern part. Moreover, ref. [80] estimated that the discharge in Chari/Logone had reduced from about 40 $km^3$/year in the 1960s to about 10–15 $km^3$ in the 1980s. This established the critical period of this drought; the Chari/Logone at N'Djamena (548,747 $km^2$) recorded 576 $m^3$/s in 1973, 236.3 $m^3$/s in 1984, 313 $m^3$/s in 1987 and 390 $m^3$/s in 1990. The year 1984 was dry and weak rainfall was recorded in the Ouham basin (Figure 2); this had an evident impact on the discharge of Chari at Ndjamena (Figure 6). However, the start of the 1990s coincided with an increase in flow towards the lake. Regarding the discharges of the Ouham and Chari rivers in the 1953–1994 period, we found a high linear correlation with $R^2 = 0.72$ (Figure 12), evidencing the impact of drought on the discharges of rivers into the whole LCB.

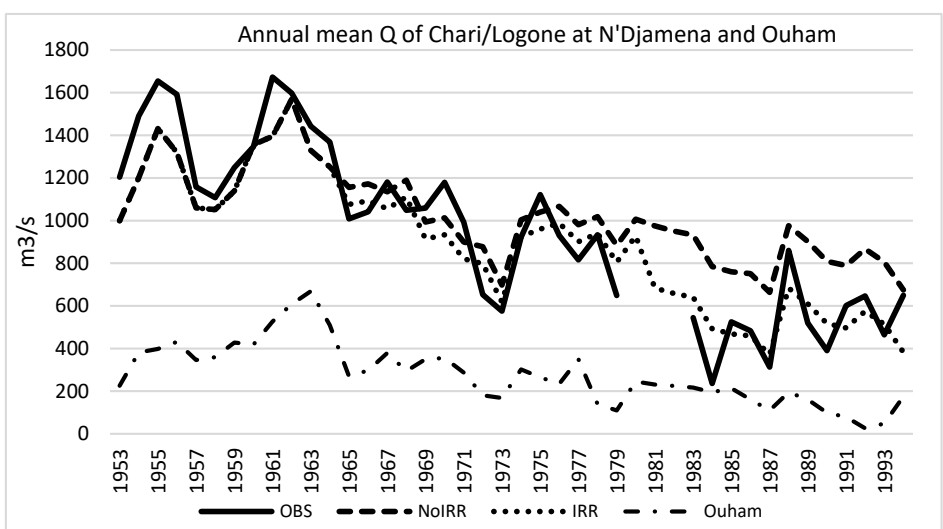

**Figure 11.** Annual mean Q of Chari/Logone at N'Djamena, according to [17] and Ouham at Batangafo. OBS: observed discharge; NoIRR: simulation no irrigation; IRR: simulation with irrigation.

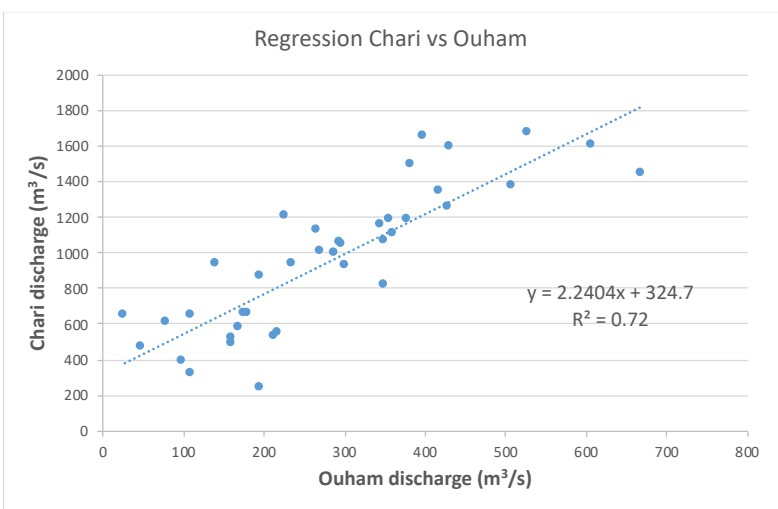

**Figure 12.** Linear correlation between the Chari and Ouham discharges (1953–1994).

Otherwise, the recent increase in the level of Lake Chad and its groundwater [42,81–86] is a consequence of rainfall reprise [42,81–89]. Despite the increasing trend in rainfall, ref. [90] noted a high interannual variability in the Sahel affecting water level and the surface of Lake Chad from one year to another.

According to [42], the average flow in the Chari–Logone basin is 42 mm/an, representing only 5% of rainfall in the period 1960–2015. Some trends between wetness and dryness are also observed in the Ouham River Basin between 1971 and 1989, with a rainfall recovery in the 1990s that induced an increasing flow at Batangafo and Bossangoa. However, ref. [84] observed that the reduction in Q in the Chari/Logone is due to human activities rather than to the impact of climatic variability; they estimated that 80% of this drop in flow is due to anthropogenic activities (water abstraction upstream of the lake). Nevertheless, these recent works have led to the reconstitution of water resources in the Lake Chad basin to be considered, as well as the fact that the Ouham River is the main contributor. Indeed, ref. [91] thinks that "the near future of Lake Chad essentially so depends on the apports of Chari, understood the Ouham River, its principal water feeder".

## 5. Conclusions

The hydrological regimes of the Ouham (rainfall and discharge), between wetness, normal or average, and/or drought periods, integrate water resource states in its basin, and they increase, fluctuate, or reduce in space and in time. Thus, the fluctuations in rainfall and discharge established the degradation of water resources in the Ouham basin through its three outlets in about four decades. Using the SPI approach, it appears that moderate to normal drought impacts rainfall in the Ouham River basin, confirming trends observed using the rainfall index. The effect of this drought was noted in important hydrological deficits, the irregularity of hydroclimatic regimes in the Ouham River, various breaks/ruptures in the annual mean Q compared to rainfall, and a decline in the mobilized volume of aquifers per outlet. These hydrological regimes were also marked by the climatic drought that started between 1968 and 1971 in the basin and its sub-basins. This phenomenon persists, with climatic warming activating the evaporating power of air, although the depletion started over the Ouham between 1963 and 1964. For weak average rainfall reductions per sub-basin or in the basin (−5%), hydrological deficits are marked in diverse degrees, despite a (very) short rainfall reprise (1991–1995). We deduce that rainfall modifications are weak per sub-basin and also in the Ubangi basin at Bangui. However, they are elevated in the annual mean discharges per outlet. In addition, various ruptures before 1970 or since 1971, as well as the annual mean Q as the rainfall per sub-basin, show homogeneous hydroclimatic periods, which express or do not express the severity of (hydro)-climatic fluctuations in the functioning of the Ouham River. With four ruptures in the Ubangi's annual mean Q (1959, 1971, 1982, and 2014, supposed but not occurred [77]) and four others in the Ouham River's mean (1960, 1964, 1971, and 1989), we can observe the similar response of these rivers to climate aggressivity, regionalizing this impact of drought. Thus, for −5% of the rainfall deficit over its basin (1935–2015), the Ubangi River at Bangui also recorded a hydrological degradation relative severity of −3% in 1971–1982 and −22% in 1983–2013 [26], underlining the impact of drought in the 1980s. Conversely, for the rainfall exceedance and deficits in its basin (Tables 2 and 3), the decline in the annual mean Q from the Ouham at Batangafo is accentuated over time. This denotes a high reduction in water resources in the Ouham (−69% on average). It can be hypothesized that human activities (fishery, agriculture), associated with diamond/gold artisanal mining, have an impact on the Ouham course. Thus, a decrease in Q or an increase upstream–downstream of the Ouham determines the regime of the water feeding Lake Chad, and therefore the fluctuation in its surface in space and time. Indeed, a strong correlation has been established between Ouham and Chari discharges, with $R^2 = 0.72$ in the 1953–1994 period confirming this relationship. Beyond the size difference of the two basins and the lack of recent data, the second and last hydroclimatic periods of the Ubangi basin can be extrapolated to the Ouham (at Bozoum and at Batangafo). These extrapolated periods, with wetness (1951–1971) and persistent drought since 1972, respectively, fit those of the Ubangi basin well. Otherwise, the recent evolution of the Chari, downstream, and Lake Chad is able to reveal the dynamics of the Ouham since 1996, without measurements/data. Nevertheless, this gap could be filled using satellite radar altimetry measurements of water height to estimate the river discharge [27,28,83], although this method cannot replace ground-based data. Indeed, recent studies testify to the lake surface extension in this interior sea [83,85,86], and to the steady increase in its South cuvette, even confirming trends relating to rainfall reprise starting in the 1990s in the whole Ouham basin. They have contributed not only to steadying the south "small regular lake" regime but also to the return of water in the north part [18]. Although these evolutions are subjected to rainfall variability in time, nevertheless [84] discloses the impact of human activities (water abstractions), which would explain 80% of the reduction in water in Lake Chad, and not essential "climate dynamics" [17].

The mediatization of water transfer from the Ubangi River to support the water volume of Lake Chad has been a reality for more than 20 years. The main issue is determining the technical feasibility of resolving the chronic drying-up of the area. The results show that the Ouham River Basin is a good hydrological system to manage the water-feeding Lake Chad. However, this will involve the control of several external factors, such as human activities, soil land use, and planetary warming.

**Author Contributions:** Conceptualization, methodology, and writing—original draft preparation, C.R.N.; writing—review and editing, D.O. All authors have read and agreed to the published version of the manuscript.

**Funding:** This research received no external funding.

**Data Availability Statement:** The data associated with this paper are available upon demand via an electronic message to the author.

**Acknowledgments:** This work was first submitted to the Water Sciences Journal in 2018. The author acknowledges all reviewers who assessed this version.

**Conflicts of Interest:** The authors declare no conflict of interest.

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
