# Peer review of "Spatiotemporal Variability in the Hydrological Regimes and Water Resources of the Ouham River Basin at Batangafo, Central African Republic"

_geosciences, doi:10.3390/geosciences13110334_

Round 1

Reviewer 1 Report

The manuscript titled "Spatiotemporal Variability’s of the hydrological regimes and 2 water resources in the Ouham River basin at Batangafo, Central 3 African Republic " was reviewed. The aim of this manuscript is to investigate the spatiotemporal variability of hydrological regimes. Authors have used data from 1951-1995 for investigation. However, the novelty of the study is not well described.  Therefore, I suggest rejection.

Main issues:

1.The writing part is very poor. Authors should proofread the entire manuscript before submission, a native English speaker works fine.

2. The Abstract is not making any sense, all the necessary components of the abstract are inconsistent and written in broken sentences.

3. Introduction must be arranged well and rewritten for a problem statement intended to address. An explicit problem statement/ objective statement should be provided at the end of the introduction.

4. The overall manuscript seems like records presented in tables and figures from the projected climate models. The novelty part is missing, data collection and distribution institutions are already presenting the climate summaries and seasonality what is your contribution.

Extensive editing of English language required

Author Response

Dear Reviewer,

Thanks for all comments you made on our paper that I taken into account through the four main issues to improve it.

I have re-written the paper through the abstract, introduction and other sections such as data and methods, results and discussions, conclusion and bibliographical references.

I let you appreciate the new version of this paper.

Regards.

Reviewer 2 Report

The manuscript “Spatiotemporal Variability’s of the hydrological regimes and water resources in the Ouham River basin at Batangafo, Central African Republic” presents a study aiming to investigate the runoff regimes and the association with rainfall records in sub-basins of the Ouham River. Data from hydrological and meteorological stations have been used, covering roughly the second half of the 20th century.

The topic of the study is interesting with potential local and regional applications for water resources management. The methodological approach and the outcomes seem sound, however the limitations of data availability may have impacts on the quality of the results and their interpretation (this issue should be further discussed – see specific comments).

The following points can be considered for the revision of the manuscript:

-          The introduction illustrates the importance of the river under study at local and regional level. However, it suggested to expand the literature review to similar international cases, highlighting its added value and why this study would be of interest to the readership of the journal and the international scientific community.

-          Additional information must be provided regarding water regulation and/or water abstraction projects (for irrigation or other purposes) that may have been constructed within the study area throughout the examined period (or later), in order to interpret with adequate accuracy and evaluate the results, and assess the extent that the presented outcomes can be still valid.

-          The description of the study area must include information which can be directly linked to the period of study (e.g. information about the population should also cover periods before 1988), and, even better, associated with specific key-points.

-          Although the ‘drought’ periods are attempted to be examined, the assessment of drought does not seem to be based on solid and objective criteria according to the state of the art on the matter. Indicatively, simple and well-established drought indices could be calculated (using the available data and easily accessible software) for the area and be considered to assess objectively the drought periods (e.g. Standardised Precipitation Index – SPI, Streamflow Drought Index – SDI).

-          The resolution of Fig. 1 is too low. The information is not readable, especially in the small (blue) map. Also, please provide the legend in English.

-          It must be discussed whether and to what extent (proper justification is needed) the available hydrometric and precipitation gauging stations can be considered representative for the conditions in the entire basin (considering the location – altitude, position within the basin -, data availability, data quality - gaps, etc.). Due to the above factors, the uncertainty level of the outcomes should be also discussed, to avoid possible misinterpretations by the readers.

-          Other minor issues: Please check for typos throughout the text. Also, check the references style according to the guide for authors.

The language is understandable. An edit by a native speaker would be suggested.

Author Response

Dear Reviewer,

Thanks for all suggested modifications you did on this paper to assist us in improving it.

Indeed we have considered it to re-write this manuscript. We did a new Fig. 1 Nevertheless we have some technical issues in applying the SDI while SPI was done.

Also about rainfall and hydrometric gauging stations we gave information's related to location, altitude etc. but we didn't get well what you suggested to be considered in this paper.

Hope on this new manuscript we will go through in finding out.

Regards.

Reviewer 3 Report

Thank you for the valuable research that you have carried out and presented in your manuscript. It was interesting to read how weather variables influence water regimes in the Ouham basin and the implications on water resources management. Here are my comments, in the hope to improve the presentation and soundness of your study:

INTRODUCTION

1. I would include a paragraph that clearly states the objectives and main purpose of your study, including the stakeholders that might benefit from your research.

METHODS

2. Because the main text is mostly in English, I would suggest to translate Figure 1 (line 103) from French to English. Also, the legend in the smaller box with the blue content is too tiny to read: please consider removing or increasing the font size. I suggest to add the latitude and longitude coordinates to the external frame of Figure 1. As example, please see Figure 1 in "Hayat, H.; Akbar, T.A.; Tahir, A.A.; Hassan, Q.K.; Dewan, A.; Irshad, M. Simulating Current and Future River-Flows in the Karakoram and Himalayan Regions of Pakistan Using Snowmelt-Runoff Model and RCP Scenarios. Water 2019, 11, 761. https://doi.org/10.3390/w11040761".

3. In section "2.2.1. R and Q Data" (line 156): Were data publicly available online for download? What was the format of the data obtained? Please consider adding references and citations on where to obtain the data to repeat your study.

RESULTS

4. Consider merging Figures 2a-c in one Figure with panels. Same for Figures 3a-c, Figures 6a-c, and Figures 7a-c.

5. In Figure 5, the Batangafo and Bozoum lines are represented by the same line type (color and texture), which makes very difficult to tell them apart. Please consider modifying the legend to facilitate reading this figure. This is also true for Figure 9.

6. In Figures 7a-c, the secondary vertical axis on the right shows groundwater units in cubic km. I suspect these units to be inaccurate as groundwater discharge (Figure 7b) is usually indicated in cubic meters per second. Please provide some guidance on this matter.

7. Line 435: In this subchapter, Figure 7 is referred to several times in support of your results. I would suggest to more precisely indicate which of the three graphs in Figure 7 you are referring to (i.e., a, b, c).

8. The unit and labeling on the y axis of Figure 9 is unclear. Please add more information on this regard. 

REFERENCES

Line 723: There is a reference that should be on a new line: "Nguimalet C. R. et D. Orange (2020). Hydroclimatic variability 723 in Tomi at Sibut, Gribingui at Kaga-Bandoro and Fafa at Bouca basins, in the Central African Republic. In: Zhongbo Yu et al., Ed 724 Sc : Hydrological Processes and Water Security in a Changing World, 2020, Proc. IAHS, 383, 79-84." 

Concluding, I suggest to revise the writing to correct some English mistakes and simplify wording throughout the manuscript as it was sometimes a bit challenging to understand the phrasing. 

Thank you

The way the sentences are constructed is a bit difficult and rough. There are some spelling mistakes and the use of words that have a different meaning from what supposedly intended. Examples might include, but are not limited to:

Line 23: "These results establish water resources decline of the Ouham River, which is the main branch of the Chari River, which should observe in the (current) surface of the Lake Chad that its feeds." Please rephrase avoiding nesting clauses.

Line 178: "For missing data ponctually, the monthly reconstitution of month i was done by the 178 averaging method of some months i of the two year...." The "i" should be in italic because they are indices and not in-text words.

Author Response

Dear Reviewer,

Thanks for your comments leading us to improve this paper.

Indeed all paper sections have been improved regarding to your suggestions; introduction, methods, results and references. Also a new Fig. 1 was made.

Regards.

Reviewer 4 Report

REPORT

The article is generally well written and well structured. The results are interesting for this region. They deserve to be published. Nevertheless, before its publication, corrections are necessary.

I.           Introduction

1. The first two paragraphs of the introduction are part of point 2.1. (The Ouham River at Batangafo basin and its sub-basins).

2. Please, reorganize the 3rd and 4th paragraphs starting first with the literature review on rainfall in the region and then on river flows.

Data and Methods

Figure 1

- Add parallels and longitudes on the map.

- Add the map of Africa in which the country is represented in mortise. International readers do not know the location of the country in Africa.

- Translate the map legend into English.

Point 2.2.2

This point should be abbreviated and inserted into the previous point. It is therefore important to delete tables 2 and 3.

3. Results and Discussion

Delete the word Discussion from the title.

Author Response

Dear Reviewer,

I acknowledge all comments you did to help us in improving this paper.

Indeed all paper sections such as introduction, data and methods, results and discussions, even the Fig. 1, were revised.

Also references were improved.

You will appreciate through the new version of the paper.

Regards.

Round 2

Reviewer 1 Report

The manuscript is improved. Accept in present form.

Author Response

Dear Reviewer,

Thanks again for your comments and suggestions for improving our paper.

We appreciated that mainly you required us to improve several sections of our paper. So we did it and also improved the English language in the paper.

Kind regards,

Reviewer 3 Report

Thank you for implementing some of my comments into the manuscript. I still suggest to include would include a paragraph that clearly states the objectives and main purpose of your study, including the stakeholders that might benefit from your research. My main comment remains the presentation of the results, especially the use of colors and patterns for lines and the legend should be more carefully chosen to best read the figures. 

In the new sections added, I could still notice spelling mistakes such as in line 86: His should be corrected to This. Please proof read your copy before submitting.

Author Response

Dear Reviewer,

We acknowledged your comments for this second round of revisions on our manuscript.

As you expected, the last paragraph of the paper was revised stating clearly the objectives and main purpose of our study. But for the stakeholders benefiting our research, we did'nt because we didn't see clear about what you suggest.

In addition, we did our the best to manage the figures legend that you will appreciate.

Furthermore; the English language was improved.

Hope we did the necessary regarding all your comments.

Regards,